# Development and Inter-Laboratory Validation of Diagnostics Panel for Detection of Biothreat Bacteria Based on MOL-PCR Assay

**DOI:** 10.3390/microorganisms9010038

**Published:** 2020-12-24

**Authors:** Pavlina Jelinkova, Jakub Hrdy, Jirina Markova, Jiri Dresler, Petr Pajer, Oto Pavlis, Pavel Branich, Gabriela Borilova, Marketa Reichelova, Vladimir Babak, Nikol Reslova, Petr Kralik

**Affiliations:** 1Department of Microbiology and Antimicrobial Resistance, Veterinary Research Institute, Hudcova 296/70, 621 00 Brno, Czech Republic; hrdy@vri.cz (J.H.); markova@vri.cz (J.M.); reichelova@vri.cz (M.R.); babak@vri.cz (V.B.); 2Department of Experimental Biology, Faculty of Science, Masaryk University, Kamenice 753/5, 625 00 Brno, Czech Republic; 3Military Health Institute, Military Medical Agency, Tychonova 1, 160 01 Prague 6, Czech Republic; jiri.dresler@gmail.com (J.D.); petr.pajer@img.cas.cz (P.P.); oto.pavlis@email.cz (O.P.); 4Military Veterinary Institute, Opavska 29, 748 01 Hlucin, Czech Republic; japonsko@centrum.cz; 5Department of Meat Hygiene and Technology, Faculty of Veterinary Hygiene and Ecology, University of Veterinary and Pharmaceutical Sciences Brno, Palackeho tr. 1946/1, 612 42 Brno, Czech Republic; gabriela.boril@seznam.cz (G.B.); kralikp@gmail.com (P.K.); 6Collection of Animal Pathogenic Microorganisms, Department of Bacteriology, Veterinary Research Institute, Hudcova 296/70, 621 00 Brno, Czech Republic; 7Department of Botany and Zoology, Faculty of Science, Masaryk University, Kotlarska 2, 611 37 Brno, Czech Republic; nikol.r@seznam.cz

**Keywords:** MOL-PCR, biothreat bacteria, magnetic bead, bioterrorism, detection panel

## Abstract

Early detection of biohazardous bacteria that can be misused as biological weapons is one of the most important measures to prevent the spread and outbreak of biological warfare. For this reason, many instrument platforms need to be introduced into operation in the field of biological warfare detection. Therefore the purpose of this study is to establish a new detection panel for biothreat bacteria (*Bacillus anthracis*, *Yersinia pestis*, *Francisella tularensis,* and *Brucella* spp.) and confirm it by collaborative validation by using a multiplex oligonucleotide ligation followed by polymerase chain reaction and hybridization to microspheres by MagPix detection platform (MOL-PCR). Appropriate specific sequences in bacterial DNA were selected and tested to assemble the detection panel, and MOLigo probes (short specific oligonucleotides) were designed to show no cross-reactivity when tested between bacteria and to decrease the background signal measurement on the MagPix platform. During testing, sensitivity was assessed for all target bacteria using serially diluted DNA and was determined to be at least 0.5 ng/µL. For use as a diagnostic kit and easier handling, the storage stability of ligation premixes (MOLigo probe mixes) was tested. This highly multiplex method can be used for rapid screening to prevent outbreaks arising from the use of bacterial strains for bioterrorism, because time of analysis take under 4 h.

## 1. Introduction

Bioterrorism refers to the deliberate abuse of pathogenic microorganisms (bacteria, viruses or their toxins) to spread life threatening diseases on a large scale and thus devastating the population of the area. Bacterial strains can be misused as biological weapons not only as a threat to human health but also in terms of agricultural abuse and ecotoxicological risks [1]. The use of biological agents in comparison with conventional weapons is very attractive to terrorists because of their relatively low cost and relative availability [2]. The most efficient way for delivery of biological agents is in the form of aerosols. However, there are a number of other ways in which biological agents can be disseminated, e.g., via food, feed, or water contamination [3]. *Bacillus anthracis*—anthrax, *Yersinia pestis*—plague, *Francisella tularensis*—tularemia or *Brucella* spp.—brucellosis are among the most abused bacteria in biological weapons [4]. The Centre for Disease Control and Prevention (CDC) classifies biological agents into categories A–C, where the category A represents the most dangerous pathogens. All the biothreat bacteria listed above belong to group A, except for *Brucella* spp., which is classified in category B [5].

*B. anthracis* is a spore-forming bacterium and is classified as one of the most important pathogens for abuse as a bioterrorist weapon [6]. The respiratory route of infection caused by inhalation of spores is a minor issue in the context of global human anthrax cases but a serious issue when associated with bioterrorism [7]. Anthrax containing letters from October 2001 confirmed that only a small amount of *B. anthracis* spores are sufficient for an outbreak due to a terrorist action [8]. The infectious dose ranges from less than 10 to over 10,000 spores depending on factors like the route of the infection or the health status of the exposed individual [7,8,9,10]. Another destructive disease is the plague caused by *Y. pestis*. Person to person transmission of *Y. pestis*, i.e., primary pneumonic plague, is possible by way of airborne droplets or aerosols and if untreated, it is 100% fatal [11,12]. A short incubation period, very low infectious dose requirement of only 1 to 10 organisms, and the progressive nature of the infection classifies this bacterium into group A of pathogens at risk of bioterrorism abuse [13,14]. *F. tularensis* is able to cause a highly infectious disease called tularemia by as little as a few microbes aspirated from the surrounding air. Tularemia is a disease of wild animals (rodents, hares, and rabbits) that can be transmitted to humans [15]. *F. tularensis* is classified into four subspecies, but only two of them are the causative agents of disease in humans. Type A is subsp. *tularensis* (predominantly found in North America), and type B is subsp. *holartica* (predominantly found in Eurasia) [16]. *Brucella* spp. causes a serious contagious disease transmissible to humans, which results in reproductive failure of infected animals [17]. Genus *Brucella* includes several species exhibiting host adaptations. *B. abortus* (cattle), *B. melitensis* (sheep and goats), and *B. suis* (pigs) belong to the most common and virulent types not only for livestock but also for wildlife and humans [18,19]. Transmission of 100 to 1000 cells are sufficient for the development of the disease [20]. Although brucellosis has been eradicated in most developed countries, it is still found in many developing countries [21].

In order to prevent the spread of the agent after a bioterrorist attack and to apply efficient preventive measures to stop the spread of infectious diseases, it is important to quickly identify and specify bacterial species [22]. The method of choice for fast and reliable identification of pathogens in various matrices is polymerase chain reaction (PCR) [23]. However, analysis of numerous samples for the presence of multiple infectious agents by PCR requires modifications of the PCR workflow. One of the possibilities is to use PCR as a suspension arrays in which the fluorescently labelled PCR product is visualized by its attachment to a specific bead. A various modifications of suspension arrays exists [24]. The latter approach, multiple oligonucleotide ligation PCR (MOL-PCR), allows the use of only a single pair of universal primers which makes optimization of the whole assay easier. The first step is then accomplished by ligation of specific targets sequence which will create a template with specific complementary primer sequences for annealing of the universal primers. The PCR products can be labelled either by the post-PCR conjugation of streptavidin-phycoerythrin (SAPE) complex with e.g., biotinylated primer or by direct labelling of a PCR primer with fluorescent dyes (Alexa Fluor532, BODIPY-TMRX). The last step is the hybridization of the PCR product using a specific 24 base DNA sequence, which is already part of the probe and its complementary sequence located on the selected set of magnetic microspheres. Precise optimization of the MOL-PCR assay is crucial and key points in the optimization process were experimentally identified [25]. Since that time, MOL-PCR has been successfully adopted for parallel detection of many human, animal, or even insect pathogens [25,26,27,28,29,30,31]. All proposed detection panels show high specificity and sensitivity and are useful for screening a wide spectrum of samples in general.

To address the need of security bodies to be capable of simultaneously detecting the four major biothreat bacteria (*B. anthracis*, *Brucella* spp., *Y. pestis* and *F. tularensis*), a comprehensive and specific protocol for the MOL-PCR suspension array was developed in this study. The scheme of the whole MOL-PCR reaction is figuratively described in the our previous study [25]. The main aim of the study was to construct a panel of detection markers for all listed pathogens using at least two DNA markers (chromosomal markers and virulence genes for each pathogen). The whole multiplex detection panel comprised an internal control (IC) to exclude inhibition of the PCR by impurities potentially present in the sample. All the diagnostic parameters including analytical specificity (inclusivity and exclusivity) and sensitivity (limit of detection (LoD)) were determined and the multiplex panel was validated according to the FDA Guidelines for the Validation of Analytical Methods for the Detection of Microbial Pathogens in Foods and Feeds [32] by a ring trial among four laboratories to prove its diagnostic potential in routine practice.

## 2. Materials and Methods

### 2.1. Bacterial Strains and DNA Extraction

*Bacillus anthracis*, *Francisella tularensis* subsp. *tularensis*, *Francisella tularensis* subsp. *holartica*, *Brucella abortus*, *Brucella suis* and *Brucella melitensis* obtained from the Collection of Animal Pathogenic Microorganisms (CAPM) at the Veterinary Research Institute (Brno, Czech Republic) were used for testing inclusivity. *Yersinia pestis* was purchased from the National Collection of Type Cultures—NCTC (Public Health England, Salisbury, United Kingdom) and was also used for inclusivity testing. Other bacterial DNA from *Staphylococcus aureus*, *Listeria monocytogenes*, *Bacillus cereus*, *Yersinia enterocolitica*, *Campylobacter jejuni*, *Salmonella enterica*, *Escherichia coli* O26, *Escherichia coli* O157, *Enterococcus faecalis*, *Enterococcus faecium*, *Vibrio parahaemoliticus*, *Yersinia pseudotuberculosis*, *Clostridium tetani*, and *Clostridium botulinum* were also obtained from the CAPM and other collections and were used for testing exclusivity (Table 1). Genomic DNA (gDNA) was purified using the DNeasy^®^ Blood & Tissue kit (Qiagen, Germany) according to the manufacturer’s protocol with a few modifications as described previously [33]. DNA concentrations were determined spectrophotometrically using a NanoDrop™ 2000/2000c Spectrophotometer (Thermo Fisher Scientific, Waltham, MA, USA) and diluted in sterile distilled water to the required concentrations, if appropriate. To test exclusivity and long-term storage effect of the ligation mix, DNA was diluted to a concentration of 10 ng/µL.

### 2.2. Internal Control

An internal control (IC) was added to each reaction to differentiate a false negative from truly negative results because of the inhibition of the MOL-PCR reaction. The IC was designed as a synthetic sequence based on the joined mitochondrial DNA sequences of two extinct species, Thylacine (*Thylacinus cynocephalus*, GenBank Acc. No. FJ515781.1) and Moa bird (*Dinornis struthoides*, GenBank Acc. No. AY326187.1). This 150 bp control synthetic sequence was synthesized de novo and cloned into a plasmid [34]. The essence of internal control is to be positive in all samples as well as in negative controls without a template—no template controls (NTC). It confirms that all reaction steps have taken place correctly.

### 2.3. Design of MOLigo Probes (Short Specific Oligonucleotides)

Pathogen-specific sequences (specific genes for detection of bacteria) were selected according to previously published data. The References are listened in Table 2. Most previous work, of which there were specific targets chosen, were based on qPCR method. Sequences of each species were extracted from the National Centre for Biotechnology Information (NCBI) database. For detection of a specific target sequence are designed MOLigo probes. Each pair of MOLigo probes is specific for a particular target sequence, but all MOLigo pairs contain the same sequence for annealing the universal primers (Reverse—Rw and Forward—Fw). One of the MOLigo probes also contains the unique 24 base DNA sequence called an xTAG (from Luminex corporation; https://www.luminexcorp.com/magplex-tag-microspheres/), by which PCR products hybridize to a magnetic microsphere with a covalently linked anti-TAG sequence. The specific parts of MOLigo probe sequences (specific sequence for a given target—part 1 and part 2) were tested with OligoAnalyzer 3.1 tool (https://eu.idtdna.com/calc/analyzer) to identify properties such as melting temperature, hairpins, dimer formation etc., of the individual target sequences used for probe design. Optimal probe sequences were finally checked by NCBI BLAST for possible non-target interactions. A more detailed scheme for designing probes for the MOL-PCR reaction is shown in Figure 1. MOLigo probe synthesis was performed by standard desalination purification (Generi-Biotech, Czech Republic). The final size of the ligation product ranged from 102 to 113 base pairs. All other parameters for MOLigo probe design were used according to the previous study [25].

### 2.4. MOL-PCR Assay

#### 2.4.1. Coating the Magnetic Microspheres with Anti-xTAG Oligonucleotides

Magplex^®^—C Microspheres sets (6.5 µm magnetic, carboxylated and internally labeled with two fluorescent dyes for bead identification) were purchased from Luminex Corporation. Each set of microspheres (12.5 × 10^6^ microspheres/mL; Luminex Corp., Texas, USA) was vortexed and 200 µL was collected into a protein low bind DNA tube (Eppendorf, Germany). The tubes were placed on a DynaMag-2 magnetic separator (Thermo Fisher Scientific, Waltham, MA, USA) for approximately 1–2 min and then the supernatant was aspirated. The magnetic bead pellet was suspended in 22 µL of 0.1M MES buffer pH 4.5 (Sigma-Aldrich, St. Louis, MO, USA) and further vortexed and sonicated for 30 s in an ultrasonic cleaning bath (Desen Precision Instruments, Fuzhou, China). Subsequently, 2 µL of 100 µM antiTAG (C6-amino modifications at the 5′-end, specific modification to form a covalent bond with the surface of the microspheres) was added into the appropriate tube with beads and vortexed. After that, 1.25 µL of a freshly prepared 10 mg/mL 1-ethyl-3-(3-dimethylaminopropyl) carbodiimide solution (solved in ultrapure H_2_O) (Thermo Fisher Scientific) was added and tubes were incubated in the dark at room temperature for 30 min. This step was performed twice with a fresh preparation of the 1-ethyl-3-(3-dimethylaminopropyl) carbodiimide solution. The tubes were vortexed every 10 min during this time. Afterwards, the beads were washed with 1 mL of 0.02% Tween 20 (Alpha Diagnostic, Texas, USA) and 1 mL of 0.1% SDS (Sigma-Aldrich) and resuspended in 40 µL of TE buffer (pH 8.0; SERVA, Heidelberg, Germany) with vortexing and removal of the supernatant using a magnetic separator after each step. The coated beads were then stored in the dark at 4 °C. To validate bead coating, direct hybridization using xTAG oligonucleotide sequences fluorescently labelled with Bodipy dye (validation TAGs) was performed using the hybridization protocol described below. The signal intensity using validation tags and coated beads was around 1000–1200 median fluorescent intensity (MFI). Prior to any downstream reaction, the real number of beads was determined by enumeration on a hemocytometer and adjusted to 40,000 beads/µL.

#### 2.4.2. Multiplex Oligonucleotide Ligation

Each multiplex ligation mix included 5 nM individual MOLigo pair probes (Table 2) with 2.5 μL10X Hifi Taq DNA ligase reaction buffer, 0.5 μL Hifi Taq DNA ligase (New England BioLabs, Massachusetts, USA), 0.1 μL of 0.05 ng/μL IC cloned into the plasmid and 2.5 μL template DNA. The reaction was brought to a final volume of 25 μL with H2O. Ligation protocol in the thermocycler (DNA Engine Dyad, Bio-Rad, Foster City, CA, USA) was set to: 10 min of denaturation at 95 °C followed by 20 cycles of 30 s at 95 °C and 1 min at 59 °C. Ligation products were stored at 10 °C until the next step of the MOL-PCR reaction.

#### 2.4.3. Singleplex PCR

Amplification of the ligation products was performed in a final volume of 24 μL which consisted of 12 μL 2X EliZyme HS Robust MIX (Elisabeth Pharmacon, Brno-Židenice, Czech Republic), 0.0625 μM universal FW primer and 0.25 μM BODIPY-TMRX-labelled REV primer (Table 2). Ligation products were added to the already prepared tubes with PCR mix at a ratio of 1:3, i.e., a volume of 6 μL. PCR consisted of initial denaturation at 95 °C for 2 min, followed by 40 cycles of 95 °C for 15 s, 60 °C for 15 s, and 72 °C for 15 s.

#### 2.4.4. Hybridization and MAGPIX Analysis

Hybridization of amplified PCR products to microspheres mediated by an xTAG sequence inside the amplified DNA and a complementary antixTAG sequence covalently bound to the surface of the magnetic beads, was performed in a bead mix formed by 1250 beads for each target (Table 2) per 1 sample; 2.5 µL of 800 mM NaCl (VWR, Stříbrná Skalice, Czech Republic) and 0.8 µL of 50 mM β-(N-Morpholino)ethanesulphonic acid (MES) monohydrate free acid ≥99%, ultrapure buffer (VWR) and adjusted to a final volume of 5 µL with 1xTE buffer at pH 8.0 (VWR). About 5 µL of the bead mix was pipetted into 0.2 mL tear-off strips (BIOplastics, Landgraaf, The Netherlands) followed by the addition of 10 µL of the PCR products from the previous step. The prepared strips with bead mix and PCR products were run on the thermocycler with the following protocol: denaturation 96 °C for 90 s, followed by 37 °C for 30 min and hold at 37 °C until further processing. After hybridization was completed, 45 µL of analysis buffer was added (10 mM Tris-Cl (pH 8.0); 0.1 mM EDTA; 90 mM NaCl and 0.2% Tween 20). Strips with hybridization products and analysis buffer were placed on a 0.2 mL Multi Rack bench (Bioplastics) and analyzed in a MAGPIX instrument (Bio-Plex MAGPIX from Bio-Rad) using xPONENT 4.2. ^®^ SOFTWARE (Luminex, Austin, TX, USA).

### 2.5. Specificity and Sensitivity of MOL-PCR Assay

The specificity of the developed MOL-PCR panel was tested using selected pathogens (Table 1). A concentration gradient (5×; 10×; 40×) of extracted DNA from bacterial strains (*B. anthracis*, *Y. pestis*, *F. tularensis* and *Brucella* spp.) was performed to confirm inclusivity. Concentrations of stock bacterial DNA ranged from 20 ng/μL (*B. anthracis* DNA) to approx. 220 ng/μL (*Y. pestis*, *F. tularensis* and *B. suis* DNA). A ten-fold serial dilution was made from the lowest concentrations (40×) and used to determine sensitivity of the liquid array detection system. The diluted DNA was analyzed in quadruplicate for reproducibility testing. The LoD was determined as the lowest concentration of sample measured at 100 median fluorescence intensity (MFI). The detection limit was determined for each pathogen as well as for the two detection markers. In the evaluation, the appropriate no template control (NTC) was subtracted for the appropriate marker.

### 2.6. Variability in Preparation of Ligation Mix and Stability during Long-Term Storage

The effect of the ligation mix composition (various MOLigo probes) and the stability during long-term storage was investigated with two independent batches of 100 premixed ligation reactions. In the first batch, ligation buffer and only MOLigo probes 1 were mixed in separate tubes and stored separately (named Ligation premix 1 = LP1); the same was done with all MOLigo probes 2 (LP2). Ligation premix 1 and 2 were mixed together just before the ligation step. Performance of this protocol was compared with all the MOLigo probes mixed and stored together in ligation buffer. Ligase was always kept separated and added to the ligation premixes just before the ligation step. *B. anthracis*, *Y. pestis*, *F. tularensis*, and *B. melitensis* DNA at 10 ng/µL were used as template in the testing and the entire MOL-PCR protocol was performed on days 0, 3, 5, 8, and 10 to investigate the freeze-thawing effect of ligation premixes. The difference in storage during freezing was compared for each of the genes by using a paired t-test, i.e., the MOLigo probes separately (LP1 and LP2) and/or the ligation mix (MIX) prepared together. The analysis was performed using Statistica 13.2 software (StatSoft Inc., Tulsa, OK, USA) and differences with *P* values < 0.05 were considered as statistically significant.

### 2.7. Inter-Laboratory Validation of the Biothreat Detection Panel

Master stocks of all chemicals were prepared, aliquoted, and frozen at the Veterinary Research Institute, Brno (VRI). Thereafter, 16 designed MOLigo pairs of probes with 1 pair of IC probes were divided into two tubes, one with MOLigo probes 1 and the second with MOLigo probes 2. Ligase was supplied separately. Pure bacterial DNA was isolated at VRI and mixed with DNA isolated from soil to mimic a natural background. Each laboratory analyzed the concentration gradient of reference DNA—*B. anthracis*, *Y. pestis*, *F. tularensis*, *B. melitensis*, *B. suis*, *B. abortus* (4 samples for each pathogen—stock solution; 5×; 10×; 40×), non-target DNA (Excess) from *S. aureus* at 20 ng/uL concentrations and 4 NTC– water only. The prepared aliquots were shipped on dry ice to partner laboratories at University of Veterinary and Pharmaceutical Sciences, Brno, Military Veterinary Institute, Hlucin and Department of Biological Protection, Techonin. The raw data from testing in all laboratories were exported as Excel files. Each sample was analyzed in biological duplicate and technical duplicate.

### 2.8. Data Analysis and Interpretation

MFI values obtained from analysis of at least 50 microspheres of each target sequence per sample were used for test evaluations. NTC with no target DNA was used to determine the background MFI value for each microsphere region. The respective NTC MFI value for the microspheres region was subtracted from the measured MFI values. Samples with MFI values higher than 100 compared to background MFI were considered as positive.

## 3. Results and Discussion

### 3.1. Biothreat Bacteria Panel Optimization

Specific genes have been selected for four bacteria that can be primarily abused as biological weapons, namely *pagA*, *BA5345* (both *B. anthracis*), *pla*, *caf1* (*Y. pestis*), *23kDA*, *fopA* (*F. tularensis*), and *omp2a* and *bcsp31* (*Brucella* spp.). These specific sequences in bacterial pathogens have been selected according to previous studies based on PCR methods, in particular real-time PCR. Relative references are listed in Table 2. Some of these are chromosomal markers, and other genes are placed on plasmids and cause a virulence of selected pathogen. Because of the gene deletion on the plasmid, two markers were preferably used for detection to confirm more identity. The assembled multiplex MOL-PCR containing all the MOLigos freshly prepared each time were tested with pure reference DNA of *B. anthracis*, *Y. pestis*, *F. tularensis,* and *Brucella* species (Figure 2) and unrelated bacterial species (Figure 3). The difference between positive and NTC samples was demonstrated when the threshold 100 MFI was applied. At the same time, the IC of MOL-PCR reaction was between 600 and 1000 MFI which shows low level of cross-reaction in amplification between target and IC. Specificity, particularly inclusivity (Figure 2) and exclusivity (Figure 3), did not show any cross-reactions or false positive or false negative interactions among all four bacterial species (Table 1). A similar detection panel for biothreat bacteria has already been assembled by Deshpande et al. 2010 [26]. Compared to Deshpande et al. study [26], *Brucella* species detection MOLigo probes and IC were added to reveal the reaction inhibition. The inclusion of IC is necessary in multiplexing for detection in diagnostic PCR assays to detect false negative results [44]. Frequently as prevention of false positive results is the most commonly used uracil DNA glycosylase [45]. A huge advantage of this method is its multiplexity and detection of a large number of pathogens in one reaction and therefore it is possible to extend the panel with other pathogens if necessary. Additionally, each bacterium was detected by two independent targets. Based on the previous findings, BODIPY-TMRX fluorescent dye was used for labelling the hybridization product instead of SAPE (time consuming) or Alexa Fluor 532 [25]. This approach enabled the omission of the separate step of PCR product labelling by phycoerythrin since BODIPY-TMRX dye successfully survives PCR amplification. As in most studies, the ligation step was separated from the PCR step to reduce the background signal and to allow better control over each step [27,43]. Compared to our previous study [25], the number of microspheres per sample was reduced to 1250 beads since this amount was sufficient to determine background and sample MFI but at reduced cost per reaction. Time generally plays a huge role in preventing the spread of the disease, epidemic or a biological warfare, so this method seems to be very suitable, as it is able to obtain results within 4 h and detect up to 50 targets and test a large number of samples at one time. All these things are important for the rapid implementation of measures to protect the population. Of course, the qPCR method (reaction time approximately 1.5 h) is most often used as the “gold standard” for the detection of bacteria, but unfortunately this method does not offer such a huge possibility of multiplexity in the same time.

### 3.2. Sensitivity Test of the MOL-PCR Reaction

MOL-PCR sensitivity was determined using ten-fold serial dilution of *B. anthracis*, *Y. pestis*, *F. tularensis,* and *B. melitensis* DNA (Figure 4). LoD for *B. anthracis* was 0.5 ng/μL for *BA5345* or 0.25–0.05 ng/μL for *pagA*. In the case of *Y. pestis*, *pla* target sequence could be detected at 0.0005 ng/μL and detection limit for *caf* ranged between 0.5 and 0.05 ng/μL. In the case of *F. tularensis*, *fopA* could be detected at 0.44–0.044 ng/μL and 23*kDA* had a detection limit of 2.2 to 0.44 ng/μL. In the case of the *Brucella melitensis*, the LoD was determined to be 0.29–0.0029 ng/μL for both target sequences (Table 3). In other studies, the LoD was determined for the Mycobacterial complex based on MOL-PCR and the detection limit was around 0.1 ng/µL. In another study, using a 13-plex for *B. anthracis* SNP typing, the LoD was 2 ng genomic DNA [27,28,29,30]. In conclusion, the biothreat panel reached a detection limit of at least 0.5 ng/uL for all pathogens and targets which corresponds with similar studies based on MOL-PCR methods. The results obtained after the evaluation of the limit of detection are that MOL-PCR method has a lower sensitivity than detection methods based on the use of DNA, thus qPCR achieves very low sensitivity. However, the MOL-PCR method is mainly a screening and semi-quantitative method, where it is very important to detect different types of pathogenic bacteria during one laboratory examination.

### 3.3. Long-Term Storage Effect of Ligation Mix

In a previous study, the effect of mixing and storing the ligation premix (MOLigo probes) was described [46]. There were up to 40 probe pairs divided into three premixes of which each premix contained 8, 5, and 3 pairs and the remaining 8 pairs were added prior to ligation mixture preparation. Based on these results, the impact of storage on background signals (MFI measured in the NTC) was confirmed [46]. For this reason, we performed two experiments (Figure 5) to compare the freezing effect of different MOLigo mix variants. In the first experiment, all MOLigo pair probes were frozen together with the ligation buffer. In the second experiment, MOLigo 1 probe and MOLigo 2 probe premixes were frozen separately. Immediately before use, premixes 1 and 2 were mixed together and ligase was added. From the point of view of stability, the separation of MOLigo probes (LP1 + LP2) resulted in better performance because none of the genes showed an increasing or decreasing trend in NTC MFI values during storage (Stability over time—trend). In contrast, the MOLigo mix method showed an increasing trend (*p* < 0.05; significance of slope of regression line) of NTC MFI values during storage of the three genes (BA *pagA*, YP *caf*, and BMspp *omp2a*). For seven of the eight genes, the MFI values for the mixed probes protocol were higher than for the LP1 + LP2 method, except for the FT *23kDA* gene. The average differences between the two protocols were statistically significant in six cases (*p* < 0.05 at least). Only the BA *pagA* and BMspp *bcsp31* genes did not show a statistically significant difference between LP1 + LP2 and mixed probes protocols (Figure 5). Our suggestion is that some undesirable cross-interactions between individual probes may occur. We therefore recommend to freeze the probe mix MOLigo 1 and MOLigo 2 separately.

### 3.4. Inter-Laboratory Validation of Biothreat Detection Panel

Inter-laboratory tests were performed to evaluate the reproducibility and robustness of the MOL-PCR biothreat panel in routine settings. Although reproducibility is defined as a quantitative parameter, it is possible to use it as a qualitative measure in validation of qPCR assays [44]. The results among all four laboratories showed agreement with one exception of one negative technical replicate of 40 × diluted sample analyzed at Military Veterinary Institute (Figure 2; Appendix A), however, the final interpretation of the results showed 100% agreement. Internal control (IC) was correct in all reactions and across all departments. The same applies to no template control (NTC) and the use of unrelated DNA (Exces) to evaluate the specificity of the probes. It was shown that the validation criteria postulated for qPCR assays by Food and Drug Administration (FDA) are applicable on MOL-PCR as well [47].

### 3.5. On-Site Usability of the Panel

To simplify the feasibility of the detection tool described here, its future conversion into a microfluidic chip or disposable cassette format is possible. A similar approach has already been introduced by Luminex corporation—ARIES^®^. This modification could not only simplify the analysis for point of care testing, but also significantly reduce the material costs and the possibility of technical error [48,49].

In conclusion, the presented protocol for the detection of four biothreat bacteria was developed and validated in this study. It connects the versatility of the MOL-PCR principle, single-tube detection of all bacteria, and increased sensitivity of detection/identification of each bacterium by at least two DNA targets. In addition, the IC of the process was included to exclude false negative results. The performance of the MOL-PCR assay was validated by a ring trial in four independent laboratories and the results showed that this assay can be implemented as a routine diagnostic for the most dangerous bacteria that could be misused as biological weapons. Of course, it is possible to enlarge the panel with detection targets for various other pathogens that could be misused as bioterrorist weapons. This method can be finished under 4 h, mainly due to its multiplexity and thus the detection (screening) of multiple amounts of pathogens in one reaction at the same time, which is indispensable in preventing bioterrorist action.

## Figures and Tables

**Figure 1 microorganisms-09-00038-f001:**
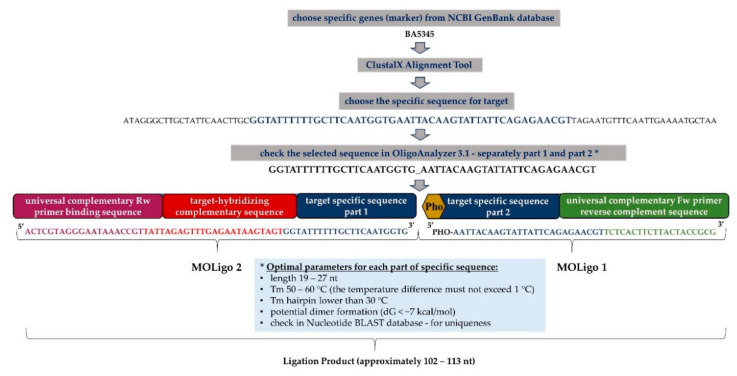
The structure of MOLigo probes demonstrated on a specific marker BA5345 for *Bacillus anthracis*: ClustalX Alignment Tool—part of a software program BioEdit, designed to perform mulTable 3. 1—analytical tool for oligonucleotides identifying their properties; MOLigo1 consists of a specific sequence for selected target (part 2) and a universal complementary forward primer (Fw)-binding sequence, MOLigo1-specific sequence is phosphorylated at its 5′end; MOLigo2 includes a specific sequence for selected target (part 1), target-hybridizing complementary sequence, which is specific for each microsphere and universal complementary reverse primer (Rw)-binding sequence. The final size of the ligation product ranges from 102 to 113 nucleotides (nt). * Optimal parameters for target specific sequences defined according to Deshpande et al. [26].

**Figure 2 microorganisms-09-00038-f002:**
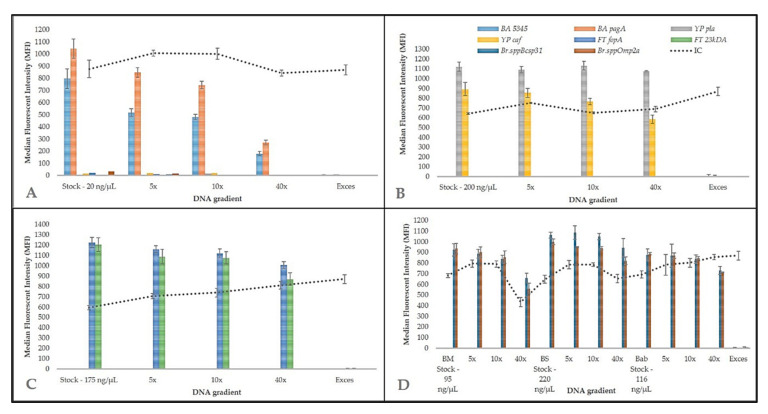
Biothreat detection panel. MOL-PCR biothreat assay for detection of (**A**) B. *anthracis*, (**B**) Y. *pestis*, (**C**) F. *tularensis*, and (**D**) *Brucella* spp. The biothreat panel was verified using DNA gra-dients from the collection of pathogenic microorganisms (CAPM) and evaluated by MFI after sub-tracting the NTC (H2O only) to the appropriate target. IC confirms that the individual reaction steps have taken place correctly. Exces (DNA from S. *aureus*) was used for confirmation as another negative control.

**Figure 3 microorganisms-09-00038-f003:**
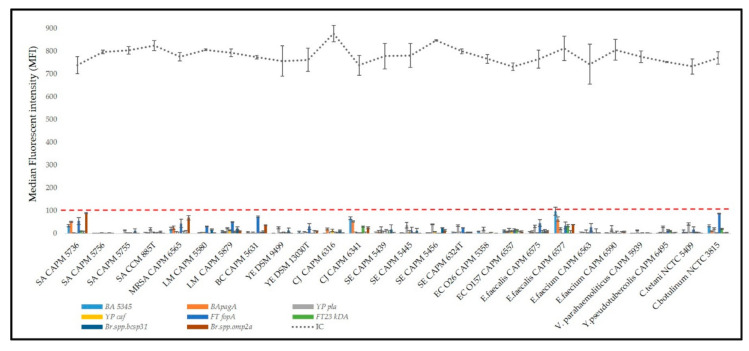
Exclusivity of biothreat detection panel based on MOL-PCR method. The exclusivity test was performed on various bacterial strains (Table 1) and evaluated as MFI. The specificity of the probes used in the biothreat panel was confirmed by the values obtained being below 100 MFI when using unrelated DNA. An IC confirmed the correct course of the MOL-PCR reaction.

**Figure 4 microorganisms-09-00038-f004:**
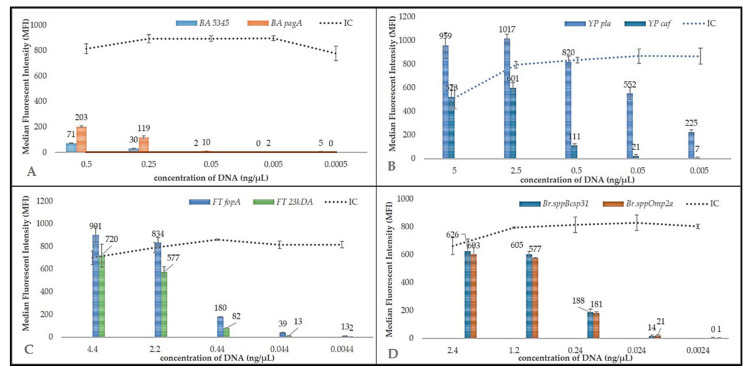
Sensitivity test.—The detection limit of the MOL-PCR system was performed in the form of a concentration gradient of isolated DNA of all bacteria and for two targets, namely: (**A**) *B. anthracis*, (**B**) *Y. pestis*, (**C**) *F. tularensis*, (**D**) *B. melitensis.*

**Figure 5 microorganisms-09-00038-f005:**
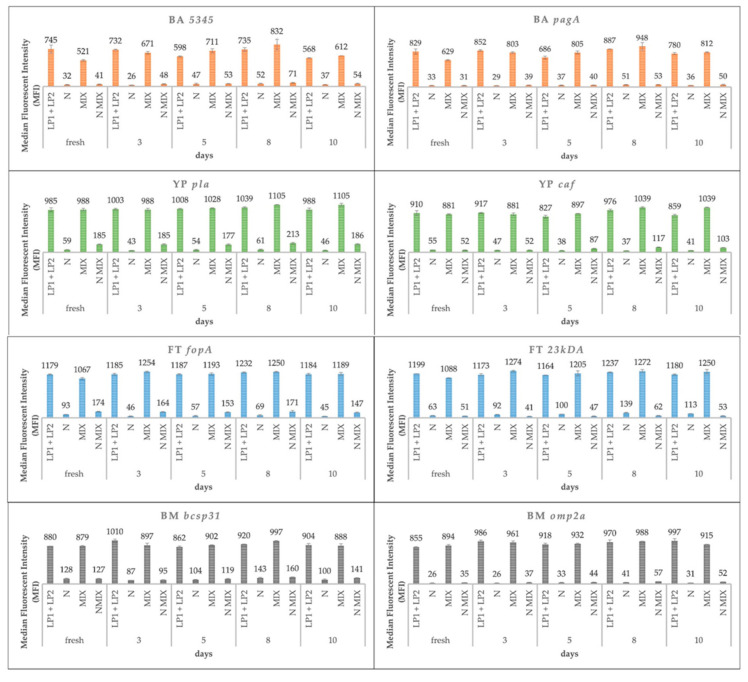
Long-term storage effect of ligation mix.—Experiment based on evaluating the stability of different types of ligation premix storage over time (MOLigo probes separately LP1 + LP2 or all MOLigo probes together—MIX). Better protocol performance was recorded when the two probes premix were stored separately. Abbreviations: LP = ligation premix; N = Negative control; MIX = mix of all probes in the ligation buffer.

**Table 1 microorganisms-09-00038-t001:** List of bacteria used in this study.

Species	Accession No.
*Bacillus anthracis*	CAPM 5001
*Yersinia pestis*	NCTC 5923 ^T^
*Francisella tularensis*	CAPM 5600
*Brucella abortus*	CAPM 5520
*Brucella suis*	CAPM 6073 ^T^
*Brucella melitensis*	CAPM 5659 ^T^
*Staphylococcus aureus*	CCM 885 ^T^
	CAPM 5736
	CAPM 5756
	CAPM 5755
Methicillin-resistant *Staphylococcus aureus*	CAPM 6565
*Listeria monocytogenes*	CAPM 5580
	CAPM 5879
*Bacillus cereus*	CAPM 5631
*Yersinia enterocolitica*	DSM 9499
*Yersinia enterocolitica*	DSM 13030 ^T^
*Campylobacter jejuni*	CAPM 6316
	CAPM 6341
*Salmonella enterica*	CAPM 5439
*Salmonella enterica*	CAPM 5445
*Salmonella enterica*	CAPM 5456
*Salmonella enterica*	CAPM 6324 ^T^
*Escherichia coli O26*	CAPM 5358
*Escherichia coli O157*	CAPM 6557
*Enterococcus faecalis*	CAPM 6575
	CAPM 6577
*Enterococcus faecium*	CAPM 6563
	CAPM 6590
*Vibrio parahaemoliticus*	CAPM 5939
*Yersinia pseudotuberculosis*	CAPM 6495
*Clostridium tetani*	NCTC 5409
*Clostridium botulinum*	NCTC 3815

Abbreviations: CAPM = Collection of Animal Pathogenic Microorganisms; CCM = Czech Collection of Microorganisms; NCTC = National Collection of Type Cultures; ^T^ = type strain, DSM = DSMZ-German Collection of Microorganisms and Cell Cultures.

**Table 2 microorganisms-09-00038-t002:** Specific MOLigo pair sequence used for detection with microspheres code and xTAG assignation (bead and xTAG numbers of unique sequences are available in the Luminex catalog).

Pathogen	Marker	Reference	Sequence 5′-3′	Size of LP	xTAG	Bead
***Bacillus anthracis***	*BA5345*	[35]	PHO-AATTACAAGTATTATTCAGAGAACGT**TCTCACTTCTTACTACCGCG** **ACTCGTAGGGAATAAACCGTtattagagtttgagaataagtagt**GGTATTTTTTGCTTCAATGGTG	112	A033	033
	*pagA*	[36]	PHO-CTGTATCAGCGGTATTTAAAC**TCTCACTTCTTACTACCGCG** **ACTCGTAGGGAATAAACCGTtgagtaagtttgtatgtttaagta**ATCTAATATCGGCATTTAATCTTG	109	A065	034
***Yersinia pestis***	*pla*	[37]	PHO-TTCTGTTGTTTTGCCTTGACATTCTCC**TCTCACTTCTTACTACCGCG** **ACTCGTAGGGAATAAACCGTgtgttatagaagttaaatgttaag**CATAATGACGGGGCGCTCA	110	A030	030
	*caf1*	[38]	PHO-AGGAACCACTAGCACATCT**TCTCACTTCTTACTACCGCG** **ACTCGTAGGGAATAAACCGTgtaagattagaagttaatgaagaa**CTTACTCTTGGCGGCTATAAAAC	106	A051	051
***Francisella tularensis***	*23kDA*	[39]	PHO-TGAGATGATAACAAGACAACAG**TCTCACTTCTTACTACCGCG** **ACTCGTAGGGAATAAACCGTgtaagagtattgaaattagtaaga**AACTAAAAAAAGGAGAATGATTATGAG	113	A066	020
	*fopA*	[40]	PHO-ACTATCTAGAAATGTTCAAGCAAGTGT**TCTCACTTCTTACTACCGCG** **ACTCGTAGGGAATAAACCGTagtaagtgttagatagtattgaat**GGGTGGTGGTCTTAAGTTTGA	112	A038	038
***Brucella spp.***	*omp2a*	[41]	PHO-CAGGCTACGAATCCAGAAA**TCTCACTTCTTACTACCGCG** **ACTCGTAGGGAATAAACCGTatttgttatgataaatgtgtagtg**CGCACTGAATCTCTGTTTTTC	104	A042	012
	*bcsp31*	[42]	PHO-TATGCCATTCGCCGCCTGA**TCTCACTTCTTACTACCGCG** **ACTCGTAGGGAATAAACCGTgtgattgaatagtagattgtttaa**CATTCTTCACATCCAGGAAACCCGAC	109	A046	046
**Internal control**	aDNA	[34]	Pho-ATTAGCACAATGAATAATCATCG**TCTCACTTCTTACTACCGCG** **ACTCGTAGGGAATAAACCGTattgtgaaagaaagagaagaaatt**TATACACACGCAATCACCAC	107	A014	036
**Forward primer**		[43]	**CGCGGTAGTAAGAAGTGAGA**			
**Reverse primer**			BODIPY-TMRX-**ACTCGTAGGGAATAAACCGT**			

Abbreviations: Pho = phosphorylation; underlined = specific sequence for target gene; bold = universal forward and reverse primer; lowercase = xTAG sequence; LP = ligation product.

**Table 3 microorganisms-09-00038-t003:** Measured limit of detection (LoD) values for the MOL-PCR method using established detection panel for the biothreat bacteria.

Pathogen	Gene (Target)	LoD (ng/µL)
*B. anthracis*	*5345*	>0.5
	*pagA*	0.25–0.05
*Y. pestis*	*pla*	<0.005
	*caf*	0.5–0.05
*F. tularensis*	*fopA*	0.44–0.044
	*23kDA*	2.2–0.44
*B. melitensis*	*bcsp31*	0.24–0.024
	*omp2a*	0.24–0.024

## Data Availability

All data underlying the results are included as part of the published article and its Appendix A.

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
