# Peer review of "Development and Inter-Laboratory Validation of Diagnostics Panel for Detection of Biothreat Bacteria Based on MOL-PCR Assay"

_microorganisms, 2020, doi:10.3390/microorganisms9010038_

Round 1

Reviewer 1 Report

 The manuscript entitled “Development and inter-laboratory validation of diagnostics panel
for detection of biothreat agents based on MOL-PCR assay
” authored by Pavlina Jelinkova,
Jakub Hrdy, Jirina Markova, Jiri Dresler, Petr Pajer, Oto Pavlis, Pavel Branich, Gabriela Borilova,
Marketa Reichelova, Vladimir Babak, Nikol Reslova and Petr Kralik presents modified version
of polymerase chain reaction (namely multiple oligonucleotide ligation PCR, in short MOLPCR) to detect bacteria that can be used as biothreat agents.
In general I liked reviewed manuscript and many of its aspects are of very high quality. I
appreciated especially test of repeatability between four laboratories. Repeatability is
extremely important and yet so often neglected aspect. Obtained results are clear,
experiments were performed in organized way and necessary controls were provided to
support presented claims.
However, few aspects of reviewed manuscript need further
improvement or more detailed explanation before publication. Thus, this manuscript
require major revision.
My main concerns can be summed up in three points:
I am not convinced about novelty of research presented in the manuscript. All shown
ideas and techniques were used before to detect almost exactly the same spectrum
of bacteria species. Described technique was already shown in Author’s previous
paper (
https://doi.org/10.1038/s41598-019-40035-5) and biothreat bacteria was
already detected with this method (doi:10.1016/j.mimet.2009.12.001).
although “Materials and methods” section is written rather precisely, few missing
crucial details hinder possibility to fully reproduce all experiments;
despite of the fact, that Authors claim to detect bacteria, in fact they are detecting
only DNA and no linkage to detection of bacteria is provided in the manuscript.
The list of issues that need to be addressed is as follows:
1. As briefly explained above, I am missing true novelty of prepared manuscript.
Although Authors mentioned few changes in comparison to already published papers
(e.g. detection of
Brucella spp, addition of Internal Control, different dye, reduced
amount of required beads per sample), all of them look rather technical.
2. To improve novelty aspect, I would like to suggest to create a linkage to actual bacteria
detection:
a. Limit of detection in ng/ml of DNA gives no information about number of
detected bacteria. Authors should provide such comparison and then all plots
should be changed to cfu/ml or bacteria/ml, which will be much more
meaningful. Since method is advertised as “bacteria detection method”, DNA
extraction should be considered as one of the steps in the detection procedure
and limit of detection in “cfu/ml” or “bacteria/ml” should be provided. This is
especially pronounced in Table 3.
b. Moreover, time of analysis should be discussed. It is not enough to simply state
that the method is “fast” (line 357).
c. In discussion section both values (LoD and time of analysis) should be
compared with methods currently used in practical applications as well as with
the best solutions described in literature. Main obtained parameters (i.e. limit
of detection and time of analysis) should be also included in the abstract.

3. Authors used few times phrase “biothreat agents” (e.g. in the title and keywords). My
suggestion is to simply use “biothreat bacteria” instead, since “biothreat agents”
contain bacteria, viruses and toxins. In reviewed manuscript only detection of bacteria
is presented.
4. I really liked description of all detected biothreat bacteria in introduction section.
However, I missed literature review about methods of their detection including
comparison of best limit of detection and times of analysis.
5. Authors used words “susceptibility”, “inclusivity” or “exclusivity” is unusual context
and in my opinion a bit confusing and unnecessary. All these things are simply either
“sensitivity” or “selectivity”.
6. Authors should avoid unexplained abbreviations throughout whole manuscript and
abstract (MOLigo in line 32, xTAG in line 153, MFI in line 245, NTC in line 240, etc).
7. It was very difficult to understand exactly how MOL-PCR works only from reviewed
manuscript. To fully understand that concept, I had to read few other articles, which
shouldn’t be the case. Here are some suggestion how to address this issue:
a. Sentence mentioned in line 205 should be placed earlier and most importantly,
figure included in citation [25] that presents idea MOL-PCR should be
presented also in this manuscript in appropriately modified and updated form.
b. Part in lines 82-94 should be expanded or rewritten to be more clear and
reader-friendly.
8. As I mentioned above, few important experimental details are missing:
a. In line 137 Authors mentioned that method to select sequences was published
before. It should be at least cited and briefly explained.
b. Error bar of every value on every plot throughout the whole manuscript always
consist of around 9.2% of main value. This is possible, yet quite improbable.
Thus, I would like to ask Authors to double check values of error bars and
elaborate a bit more about statistical approach that they used. How many
repetitions were performed in case of every experiment? What averaging do
error bars represent?
c. I had problem understanding what is the purpose of the Internal control
(section 2.2). Should it give positive signal in the performed MOL-PCR or not?
d. Figure 1 is not explanatory at all. What is ClustalX and how it was used? How
OligoAnalyzed 3.1 was used and what was the outcome? Abbrevaitions “Rw”,
“Fw”, “nt” are not explained. Why values of parameters described in blue box
were selected in that range and how Authors managed to check them? This
whole figure should be reorganized and ideally completely redesigned. Last
but not least: fonts should be bigger.
e. In line 139 names “MOLigo1” and “MOLigo2” were introduced without any
explanation. I understand that for someone working daily with MOL-PCR this
is obvious, but for non-experts that will be reading this publication, this is just
another mystical name.
f. In Table 2 Authors use name “xTAG”. More explanation would be useful. I also
don’t understand meaning of column “xTAG” in Table 2. Are those some
cataloged xTAGs, that can be found by name in some database or are those
only internal labels of xTAGs among Authors of that manuscript? If the second
case is true, this column is not necessary in the manuscript.
g. In which solvent EDC reaction was performed (line 164)?

h. What “C6-amino modifications at the 5’-end” in line 162 means?
i. More information about utilized beads would be useful (size, materials,
commercial name, coating, etc).
j. It would be nice to add to SI results of validation of coating of beads, described
in line 171-173.
9. I do not understand what is the purpose and final conclusion of results obtained for IC
in Figures 2, 3, 4, S1, S2, and S3.
10. In many Figures error bars are missing. For example: Figure 3, 5, S2B, S3A.
11. Figures 2 and 4 present basically the same experiment, simply performed for different
range of DNA concentrations. Authors should consider unifying X axis as “ng/uL” and
combining them into one plot.
12. Table 1 contain results in column “MOL-PCR detection”. This data should be included
in Result section, not Materials and Methods.
13. Manuscript should be checked again for mistakes and typos, as there are still some
present (e.g. 10^6 in line 156 or “five” instead of “5” in line 197).
14. Authors should try to explain why storage of mixed probe resulted in higher MFI
values.
15. What is “N” present in X axis of Figure 5?
16. Since Authors get very nice agreement of results from all four laboratories, Table 4 is
not necessary and should be removed. Whole information provided in Table 4 can be
summarized in sentence present in lines 338-341.
To conclude, presented idea is very interesting and can be beneficial in field of biothreat
bacteria detection. Obtained results are convincing and promising. However, few issues have
to be explained, improved and solved (clear scientific novelty, actual bacteria detection and
not just DNA detection, missing experimental details) before manuscript will be ready for
publication.

Author Response

Dear Editor and Revioewer,

here I am sending revisions to the article entitled: Development and inter-laboratory validation of diagnostics panel for detection of biothreat bacteria based on MOL-PCR assay. Revisions are documented as an attachment in word format, see. below as an annex

Kind Regards,

Pavlina Jelinkova (correspondent author), Veterinary Research Institute, Brno - Czech Republic

Reviewer 2 Report

The following comments may help the authors to improve the manuscript:

  1. The PCR is a golden standard method, currently, it appeared another method called LAMP which has the potential to replace PCR as it needs only an isothermal setup. Please elaborate on this matter as a future perspective.
  2. Can your methods be used in a point of care setting including microfluidics and lab on a chip for future application?

Author Response

here I am sending revisions to the article entitled: Development and inter-laboratory validation of diagnostics panel for detection of biothreat bacteria based on MOL-PCR assay. Revisions are documented as an attachment in word format, see. below as an annex

Kind Regards,

Pavlina Jelinkova (correspondent author), Veterinary Research Institute, Brno - Czech Republic

Round 2

Reviewer 1 Report

I am satisfied with responses of Authors to my review. Practically all of my remarks were addressed in sufficient way and manuscript was corrected and improved respectively. In my opinion manuscript is now ready for publication in MDPI Microorganisms journal.

Said that, I have one small remark that should be treated as general comment rather than criticism of current version of paper:

Ad “Response 2a”: I understand that to be able to compare obtained results with other MOL-PCR methods, Authors kept units in “ng/ul” of DNA. However, on top of that Authors could add additional recalculation and provide both versions of LOD: in ng/ul of DNA and in cfu/ml of bacteria. Such way, Authors would provide more information and their work could be compared not only with MOL-PCR methods but with all methods for bacteria detection. I am aware, that going from work with DNA to work with bacteria requires different laboratories and equipment, but I believe it can be very useful.

Author Response

Manuscript was revised according to the report 2 by R2

Reviewer 2 Report

The authors have not included the discussion inside the manuscript. The definition of point of care testing should be included when mentioned about it, see, for example, https://doi.org/10.1016/j.trac.2020.116004

Author Response

(The authors gave the same response as above.)
